# RNA-Mediated Non-Mendelian Inheritance in Mice: The Power of Memory

**DOI:** 10.3390/biom15040605

**Published:** 2025-04-21

**Authors:** Minoo Rassoulzadegan

**Affiliations:** 1Department of Medical Biology, Erciyes University, Kayseri 38039, Turkey; minoo@erciyes.edu.tr; 2Centre de Biochimie Valrose, University of Nice Sophia Antipolis, 06000 Nice, France

**Keywords:** genetic, phenotype, non-Mendelian inheritance, RNA, cell memory

## Abstract

The mouse genome is transcribed at different rates in both directions from the newly formed genome after fertilization. During embryonic genomic activation (EGA/ZGA), the first RNA metabolism creates heterogeneity between blastomeres. Indeed, ZGA-dependent maternal RNA degradation is crucial to regulate gene expression and enable the initiation and acquisition of full developmental competence. Subsequently, from the new genome, in addition to mRNAs, a wide range of regulatory ncRNAs are also transcribed. Regulatory ncRNAs (non-coding RNAs) have profoundly influenced fields ranging from developmental biology to RNA-mediated non-Mendelian inheritance, exhibiting sequence-specific functions. To date, the database cataloging ncRNA is not exhaustive, but their high sequence diversity, length and low expression level can vary within the same genome depending on environmental conditions, making understanding their functions often ambiguous. Indeed, during transcription control, cellular RNA content varies continuously. This phenomenon is observed in genetically identical organisms studied—bacteria, flies, plants and mammals—due to changes in transcription rates, and therefore, it impacts cellular memory. Importantly, experimental data regarding the simple modification of RNAs levels by microinjection into fertilized mouse eggs suggest that they certainly play a driving role in establishing and transmitting newly formed expression information. The idea here is that, even in a stable genome, transcripts can vary rapidly and significantly in response to environmental changes, initiated by transcriptional variations in the genome, thus altering cellular memory.

## 1. Concept of RNA-Mediated Inheritance

### Memories of the Species

The Mendelian heredity mediated by DNA ensures the stability of the species’ memory. This information is affected by rare and irreversible changes (1 error per 1 to 10 million cell divisions) in the structure and random distribution of chromosomes during meiosis [1]. Figure 1 shows different modes of inheritance, genetic with DNA variation and RNA-mediated non-Mendelian without.

The term non-Mendelian inheritance classically refers to modification of the genome occurring from one generation to the next and in the genome of the offspring, notably by the mobilization of transposable/retroviral elements and their insertion at a position different from that of the parents. Therefore, this causes variation in DNA. These events are also rare and continuously influence the transcription of the newly inserted region, thus leading to very distinct phenotypes between individuals due to genetic alteration by integration of transposable elements [2].

In contrast, genetically identical organisms can exhibit hereditary differences observed in bacteria [3], flies [4], plants [5] and mammals [6,7] with high efficiencies (as opposed to Mendelian and non-Mendelian differences) due to changes in transcription rates (without nucleotide changes) and thus changes in cellular memory [8,9]. This is referred to as high-efficiency RNA-mediated non-Mendelian inheritance, responsible for the appearance of highly variable phenotypes from one generation to the next, reversible without nucleotide modifications of the genome [6].

We do not use the term “epigenetics” here, as it is widely used for many purposes and primarily deploys a species-specific program. It may use the same type of mechanisms, but it differs, to some extent, from the newly acquired phenotype in terms of maintenance, inheritance and possible retention in an evolutionary process.

It is currently unknown how the parental environment can affect the transcription of genetic information in the next generation without altering the DNA sequence. It is also unknown how this newly formed information affects the phenotype and is transmitted. Several structural and molecular variations have been reported at the chromatin [9,10,11], DNA [11,12] and RNA [5,6] levels with differential methylation [13,14,15]. RNA, in particular, exhibits rapid turnover under environmental stress, allowing the initiation of radical changes and impacting immediate cellular memory capabilities.

Previously, in the early days of molecular biology research on bacteria, proteins were considered the main factors involved in the transcription process. However, non-coding RNAs were unknown and, more importantly, the techniques were not available. RNA was even been proposed as a player, but its role has not been studied in the initiation of variations and as a newly formed information [3]. The genetic risk associated with complex inherited traits involves the cumulative effects of peripheral genes on mechanistic “core genes”, suggesting the existence of a genetic network [8].

Alterations and changes in RNA levels are broad and could constitute specific initial events in response to rapid environmental changes in adaptation processes, adjusting transcription levels and rates.

## 2. Experimental Evidence

The study of information generated by regulatory ncRNAs in eukaryotes has opened important perspectives for the evaluation of genome function. ncRNAs are variably expressed in different cell types in vivo and are particularly involved in multiple regulatory networks that change depending on conditional variations [10,11].

In the case of highly efficient RNA-mediated non-Mendelian inheritance [4,5,6], with variation in transcription levels, a variable memory record is required to vary the transcription rate of the genome. This can only be rapidly ensured by RNA fragments, as occurs after fertilization to modify and vary the cellular memory of daughter cells.

RNAs not only possess sequence homology with the genome, but have unlimited possibilities for size variations, modification of the nuclear compartment, cytoplasmic transitions, and, above all, hybridization with DNA (the genome), thus creating a cellular memory [16,17]. A faithful version of genomic transcripts, with multiple possibilities for rapid variation, can therefore only be ensured by RNA. Indeed, experimental results have shown that various environmental factors have intergenerational effects [7], notably through modifications of small non-coding RNAs present in spermatozoa [14]. Furthermore, microinjection of sperm RNA or synthetic RNA into fertilized mouse eggs varied the phenotypes of mice born after manipulation and produced novel phenotypes [6,7]. Subsequently, altered DNA methylation profiles were detected in different tissues types in the next generation [13].

Transcriptional variations are also observed in heterozygotes, induced by a given gene. These changes can persist for some time and be transmitted to the next generation, even in offspring carrying a wild-type genotype [3,4,5,6]. Environmental changes also affect organisms, initiating transcriptional changes in the genome (from both alleles). Once introduced, they alter the rate of multiples endogenous transcripts and control of transcription factors that constitute the positive feedback loop in the native organism.

These transcriptional alterations directly contribute to the inheritance and gene expression profiles of future given cell types. At the same time, these alterations must/will create novel genome-specific identities. RNA could hybridize to the genome, creating a variable DNA/RNA hybrid structure. DNA–RNA hybrids are triple-stranded structures with displacement of single-stranded DNA (ssDNA). In cells, these hybrids form during transcription and play an important role in structural regions of the genome such as telomeres and centromeres. They render regions of the genome, such as the S regions of immunoglobulin (Ig) genes, unstable, thereby increasing the frequency of recombination. DNA-RNA hybrids are frequent and dynamic along the genome. In promoter regions, they influence the transcription rate. In fact, RNA molecules hybridized to DNA could interfere with the transcription rate or create a new one [16,17,18,19]. These new profiles are conserved within a specific cell type. Genome tagging with RNA is the most reliable way to record the transcriptional activity events induced in a given cell type.

Curiously, despite all these interesting findings, robust intergenerational biological effects or RNA-mediated signals have been replicated in independent cohorts, but confusion and misunderstandings persist. Moreover, as Kevin Struhl [20] pointed out, the distinction between the multiple terms epigenetics, epigenomics, etc., is often confused and misunderstood. In particular, RNA-mediated non-Mendelian inheritance in mammals is not always analyzed between generations and in a stable and homogeneous genetic context. But often, this information is ignored for various unrelated reasons. And yet, only mouse models have been able to experiment with the initiating role of RNA. This manipulation is fundamental to demonstrate RNA-mediated instructions. More precisely, it is often argued that if non-Mendelian inheritance exists in mammals, why have many studies not yet identified clear and reproducible changes linked to a specific health problem?

This remains a central argument against RNA-mediated non-Mendelian inheritance in general. However, one important point raises a question: since Mendel, we have been formatted by linear genetic factors. Indeed, the DNA sequence can confirm data with precision and certainty in the presence of phenotypic changes. This is why we continually seek such confidence in biology. This leads us to ask: does such a probability of finding the same type of certainty as in genetics exist in RNA-mediated non-Mendelian inheritances? If so, what? DNA contains information, but RNA could change the rate of transcription. Indeed, the only eligible candidate is RNA, which contains sequence information dictated by DNA, the genome, and can be sequenced and tested [19]. Moreover, it can form a specific, powerful, and faithful network with all cellular contents: proteins, RNA, and especially DNA. It remains to be discovered with what efficiency and/or precision the variable transcriptional product constituting the RNA-mediated non-Mendelian memory (RNA as initiator of transcriptional variation) is altered, modified and maintained in a given genetic environment.

On the other hand, this does not mean that the same variations are systematically found from one person to another, simply because the effects of the environment are unlimited (The origin of species, C. Darwin 1959 [21]). In each case, the impact of the factors can be different, as it depends largely on the genetics of the person and their environmental conditions. Reliable information circulates, but with certainly unlimited possibilities since RNA-mediated non-Mendelian inheritance is observed and deployed for transmission in the form of RNA fragments with possibilities of subtle changes and interactions with DNA. There are many examples of subtle variations visible to the naked eye in the living world, such as variations in flowers or skin color. Indeed, observing nature with color variations has always opened new perspectives for understanding molecular mechanisms.

## 3. Future Perspectives

RNAs forming hybrid structures with DNA can very well impose a memory on the genome. They thus create DNA/RNA hybrid zones on the genomic sequence depending on the concentration of candidate RNAs present at the time of division and subsequently influence the transcription rate. Indeed, the transmission of trait expression variations from a heterozygote to its offspring, even with a wild genotype, is a perfect example of continuous changes in gene expression levels [3,4,5,6]. Moreover, microinjection of miRNA into fertilized eggs experimentally demonstrates the induction of its variation in various tissues in mice [22].

It is important to continue to question current data, but also to appreciate models that might provide valuable additional results in such an endeavor.

A preliminary approach to resolving the controversy and clarifying its relevance to all traits is the continued creation of genetically homogeneous and stable models. For example, different rodent models of stress, trauma, and feeding have been used to demonstrate consistent behavioral and physiological adaptations in response to paternal life history, ranging from early trauma to mild and severe chronic stress in adulthood, including dietary restriction or exposure to toxic substances [23,24,25,26,27,28,29,30,31]. Indeed, through these stable mammalian models, independent research groups have demonstrated the specific role of RNA molecules in regulating mammalian development and their influence on physical and behavioral health [25,26,27,28,29]. Moreover, short and long noncoding RNAs in sperm are essential for the intergenerational transmission of altered phenotypes. Indeed, for the intergenerational transmission of effects, certain offspring phenotypes are reproduced by microinjection of non-coding RNA collected from sperm into fertilized oocytes [6]. These results from research on mouse models demonstrate that RNA-mediated non-Mendelian inheritance regulates the impact of paternal stress on offspring behavior and physiology [27]. This brings us closer to understanding its impact on human health [30], but also raises new questions. If paternal health influences our behavioral traits and mental health, it is also expected to influence other aspects of health [31].

Any variations affecting transcripts and inducing cis or trans regulatory changes generating an effect could be favorable or unfavorable to the regulation of gene function. Such changes could randomly result in a dominant trait. At the same time, transcripts could be dose-sensitive if their concentration variation disrupts another trait. Therefore, modified transcripts would exhibit more variation [11,12]. How these signals are formed and stored to remain in the cellular state remains to be clarified by further research.

In humans, too, an imbalance is expected due to various effects, while maintaining the dominant favorable effect resulting from better adaptation [29,30,31,32]. The transcriptional activities of the alleles are consistent with data from several studies indicating increased adaptation to regulatory changes. Previously reported effects of Rehostat (e.g., imprinted loci) could help explain these mysterious properties as well as other dosage effects [33]. Success will depend on the RNAs produced, which are themselves affected by the genotype.

Not all genes are dose-sensitive [17,33]. The reason for this is unknown; there are only speculations or a few examples regarding sequences (repeated motifs) or chromosomal positions. The altered expression of transcripts disrupts stoichiometric balance, and interacting factors compromise the characteristics of a given individual. If the same regulatory change has an independent beneficial effect, overall phenotypic fitness may still exceed that of the wild type in both directions. Moreover, different quantitative responses or equilibrium thresholds are possible for different traits controlled by the same gene. Loss of phenotype may occur even if considerable activity of the assay-sensitive protein persists due to regulatory changes induced by variations in DNA-targeting RNA levels [34].

Nevertheless, decisive discoveries are expected on the information transmitted by ncRNAs during their interactions with the genome and for the establishment of cellular memories [33]. This involves not only understanding species specificity [35] and how they vary according to the genome, but also elucidating their powers in a process of adaptation towards more radical modes likely to influence the directions of molecular evolution [36]. The choice and development of less artifactual technologies are necessary to understand ncRNA and their fundamental contributions to biology. The study of DNA/RNA hybrids in humans using robust and reproducible techniques is essential to pave the way for tracking unlimited variations in expression of phenotypes.

## Figures and Tables

**Figure 1 biomolecules-15-00605-f001:**
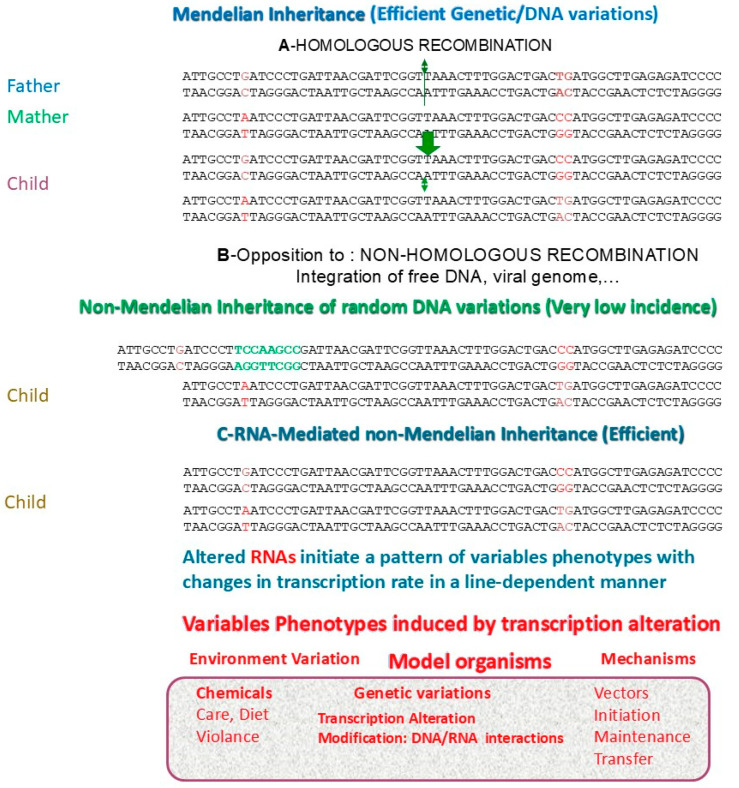
Modes of inheritance. A—Mendelian random homologous recombination between parental alleles, leading to rare DNA variations. B—Non-Mendelian recombination with non-homologous elements such as transposons or viruses, random and rare events with DNA variations in germ cells. C—Non-Mendelian inheritance with genome unchanged from the parents but with variable transcription rates.

## Data Availability

Not applicable.

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
