# Peer review of "RNA-Mediated Non-Mendelian Inheritance in Mice: The Power of Memory"

_biomolecules, 2025, doi:10.3390/biom15040605_

Round 1
Reviewer 1 Report
Comments and Suggestions for Authors
Manuscript Overview
This article explores Non-Mendelian inheritance in mammals, with a particular focus on the role of RNA in regulating gene expression and transmitting genetic information. The core argument of the paper is that small RNAs (sncRNA) and other non-coding RNAs may act as "memory carriers", influencing transcription rates in response to environmental changes and being transmitted to the next generation through sperm or other mechanisms. This mechanism challenges the traditional view that genetic inheritance is solely determined by DNA sequences, suggesting that RNA not only functions as a regulatory factor but may also play a more direct role in heredity.
This manuscript covers the following key areas:
- Background of Non-Mendelian Inheritance:
The article introduces Mendelian inheritance, which traditionally attributes heredity to stable DNA transmission. It explains how transcriptional changes across different species (bacteria, Drosophila, plants, and mammals) can lead to stable, heritable phenotypic variations. - RNA as a "Genetic Memory Carrier":
The article discusses the crucial role of non-coding RNAs (such as miRNAs and sncRNAs), which can rapidly respond to environmental changes and regulate gene expression. Various mechanisms of RNA-mediated transcriptional regulation are explored, including the formation of DNA-RNA hybrid structures, which may influence cellular memory establishment. - Experimental Evidence:
The article presents a series of animal model studies, particularly in mice, demonstrating that paternal RNA alterations can impact offspring behavior and physiological traits. Experimental injection of sperm RNA into fertilized eggs resulted in observable phenotypic changes, supporting the role of RNA in non-Mendelian inheritance. - Controversies and Challenges:
While substantial research supports RNA-mediated non-Mendelian inheritance, the article acknowledges ongoing skepticism regarding its stability and generalizability. It discusses the potential variability of RNA effects across different genetic backgrounds and environmental conditions, making it difficult to establish consistent evidence akin to DNA sequence mutations. - Future Research Directions:
The article emphasizes the need for more stable mammalian models to further investigate the mechanisms of RNA inheritance. It proposes that RNA might exert gene regulatory effects via "dose-sensitive effects," providing insights into complex inheritance phenomena.
Overall Evaluation:
This article presents a timely and compelling discussion on RNA-mediated non-Mendelian inheritance, with a clear and well-structured argument supported by extensive literature and experimental data. Its strengths include:
- Innovative Perspective: The article challenges conventional views by proposing RNA as a potential hereditary information carrier.
- Comprehensive Literature Support: It references a wide range of studies across various species, indicating that RNA-mediated inheritance is not restricted to isolated cases.
- Experimental Evidence: The inclusion of multiple mouse model experiments enhances the credibility of the hypothesis.
However, certain aspects of the article require improvement, as detailed below:
|
Issue |
Specific Revision Suggestions |
|
Unclear chapter structure |
Add clear subheadings, such as “Mechanisms of RNA-mediated inheritance,” “Experimental evidence,” and “Future perspectives” to improve logical clarity. |
|
Insufficient data support |
Include additional data on RNA level changes and illustrate their correlation with final phenotypic expression. Ideally, provide quantitative data to strengthen the argument. |
|
Lack of depth in controversy discussion |
Expand on the discussion of opposing viewpoints and elaborate on how experimental validation could further support the mechanism of RNA inheritance. |
|
Unclear terminology definitions |
Provide precise definitions of key terms in the introduction or discussion, such as “cellular memory” and “DNA-RNA hybrid structures.” |
Author Response
Thank you for your review and valuable comments. Please see the attachment for the detailed reply.

Reviewer 2 Report
Comments and Suggestions for Authors
NonMendelian inheritance in mammals: the power of memory
Minoo RASSOULZADEGAN
This MS summarizes important results on the roles of parental ncRNAs on heritable trait expression in progeny. The MS is very difficult to read. Frequently, it is not possible from this MS to know what events are taking place at the molecular and/or cellular level. Many statements are poorly or ungrammatically written. Given the importance and novelty of the phenomena referenced, it is important that this information be more clearly presented. Some detailed comments follow. They are in no way comprehensive.
Lines 6-7: Need to specify genomes of which organisms are being described.
Line 11: “high numbers and low-levels expression” should read “expression levels”
Lines 13-17: Should state a summary of what are the experimental results about sncRNAs as memory molecules, not just the conclusions. This can be done in a few words or simply by moving lines 25-27 into the abstract.
Lines 31-31: unclear what “other modifications” refers to. Are they not methylations, and do those methylations occur on DNA or RNA?
Line 34: aren’t there better ways to describe RNA-dependent transmission besides “non-Mendelian” which has many other phenomena attached to it?
Line 37: what kinds of “fragmentation” are meant, and is that the best way to describe them?
Lines 36-39: This sentence is ungrammatical, and its meanings hard to understand.
Lines 56-67: the paragraph makes some important points, but they need to be clarified. What, exactly, are the features of RNA-based inheritance that distinguish it from DNA-based inheritance? Why aren’t the effects of prior parental experience explicitly mentioned?
Lines 68-72: Again, the sentence is hard to follow, and experimental or statistical methods to deal with individual variations are not explained.
Lines 72-74: Difficult to understand. What do “vector” and “wave” mean? Are there citations fof these terms?
Lines 75-81: Define “variations.” Why do they depend upon heterozygosity of the recipient? Define the factors that modify these “variations.” What process can introduce them to the germline?
Lines 82-87: What does “a DNA/RNA hybrid structure” mean? What kind of vectors? Why no reference for statement on past results in lines 85-87?
Lines 109-110: Cite references for “preliminary results from research on mouse
models.”
Lines 123-125: What does the term “alleles” mean in this sentence. Does it refer to DNA or RNA structures?
Lines 131—136: References needed for the statements made here.
Lines 139-150: The statements here are really more appropriate for the introductory portion of the MS.
Comments on the Quality of English LanguageThe writing is hard to follow and many terms are not well defined. The overall argument needs to be presented in a more logical fashion so that readers not familiar with trans-generation ncRNA transmission can follow the arguments.
Author Response

(The authors gave the same response as above.)

Round 2
Reviewer 2 Report
Comments and Suggestions for Authors
The MS has been improved, but sections are still difficult to follow. Conceptually, the MS does not clarify the difference, if there is one, between Non-Mendelian Inheritance and RNA-directed epigenetic regulation. However. the importance of the phenomena described is so timely and relevant that there is a benefit in publishing this MS.
Comments on the Quality of English LanguageThe author should have a colleague who is a native English speaker to go over the MS and make sure the text is understandable.
Author Response
Dear Editor,
Thank you for your time and effort in reviewing this document and for the additional time you gave us to make changes. Below are my responses to the reviewers, which I hope will meet your expectations.
I submitted a new version and emailed the latest version with all changes indicated by color.
Reviewer
The MS has been improved, but sections are still difficult to follow. Conceptually, the MS does not clarify the difference, if there is one, between Non-Mendelian Inheritance and RNA-directed epigenetic regulation. However. the importance of the phenomena described is so timely and relevant that there is a benefit in publishing this MS.
Author
Thank you for your patience. I tried to be more specific. However, I disagree with your sentence:
If there is a difference between non-Mendelian inheritance and RNA-mediated epigenetic regulation…
Yes, of course. There is even a major difference: in non-Mendelian inheritance, the DNA is modified by the random insertion of a DNA fragment.
In RNA-mediated non-Mendelian inheritance, on the other hand, the genome is not modified, but the transcription rate is modulated without modification of the DNA; this is a major difference.
However, following your recommendations, I have tried to clarify this point, notably with the help of a figure.
Best regards,
Minoo
